# Unraveling Systematic Biases in Brain Segmentation: Insights from Synthetic Training

**Romain Valabregue** [1]                                           ROMAIN.VALABREGUE@UPMC.FR

**Ines Khemir** [1]                                               INES.KHEMIR@ICM-INSTITUTE.ORG

**Guillaume Auzias**[2]                                        GUILLAUME.AUZIAS@UNIV-AMU.FR

**Francois Rousseau**[3]                                 FRANCOIS.ROUSSEAU@IMT-ATLANTIQUE.FR

**Mehdi Ounissi** [1]                                           MEHDI.OUNISSI@ICM-INSTITUTE.ORG

[1] *DAC, CENIR, ARAMIS, Institut du Cerveau (ICM) - Paris Brain Institute, Inserm U 1127, CNRS UMR 7225, Sorbonne Universite, Paris, France*

[2] *Aix-Marseille Universite, CNRS, INT, UMR 7289, Marseille, France*

[3] *IMT Atlantique, LaTIM INSERM U1101, Brest, France*

**Editors:** Under Review for MIDL 2024

## Abstract

This study examines how the quality of ground truth labels affects brain MRI segmentation models. We investigate the potential of synthetic learning to mitigate systematic biases present in training labels. Through a validation on high-quality datasets, in the Putamen region, known for systematic segmentation errors like the inclusion of parts of the Claustrum, we demonstrate the effectiveness of the synthetic data approach in correcting these errors and enhancing segmentation accuracy. Our findings highlight the limitations of pseudo-ground truth labels derived from automated techniques and underscores the importance of precise, expert-validated labels for accurate, unbiased validation.

**Keywords:** deep neural network, Synthetic, brain segmentation, bias, Validation

## 1. Introduction

In this study, we investigate the impact of the definition of the ground truth labels used for training and evaluating a segmentation model for brain MRI. Access to large samples of anatomically accurate segmentation maps is key for designing machine learning models. Manually annotating high-quality labels for 3D MRI poses significant challenges, prompting researchers to resort to the use of pseudo-GT labels derived from popular automated tools like Freesurfer (Billot et al., 2023; Svanera et al., 2024; Bontempi et al., 2020; Henschel et al., 2020, 2022). Pseudo-GT labels allow to bypass the bottleneck of manual annotation, but can be affected by errors. Deep learning (DL) models are capable of generalizing and correcting errors to a certain degree, if their distribution across the training set is random. If the training set is contaminated with **systematic** errors, such as the consistent over- or underestimation of specific brain structures, DL models will learn and replicate such a bias. In this work, we investigate the potential of training a model using **synthetic** data for

which the alignment between the image intensity profile and labels boundaries are perfect by construction, to minimize learning biases from flawed labels. Our focus is on the Putamen, a brain region characterized by distinct contrast and a clear spatial configuration, which facilitates an effective visual assessment of the model's quantitative performance metrics.

## 2. Data and Methods

We consider two datasets: **MICCAI-2012 Multi-Atlas Labeling Challenge** (Landman and Warfield, 2019): We used the data from the 20 subjects of the test set defined for the challenge. The manual segmentation labels (**GTManu**) were provided by Neuromorphometrics. For this dataset, we also considered the results of the winner of the challenge **#1** (**PICSL**), which is a method based on template deformation and a patch based refinement (Wang and Yushkevich, 2013)] **HCP** (Van Essen et al., 2013; Glasser et al., 2013): We used the label maps from 40 subjects for generating the simulated data used for training the synthetic models, and another set of 80 subjects for testing. We resliced the MRIs to 1mm for better comparison with the first dataset. For both datasets, we performed whole brain segmentation with two widely used software: **FSL** (Patenaude et al., 2011), Freesurfer (**FreeS**) (Fischl, 2012). We then used our own implementation of SynthSeg (Valabregue et al., 2023) to train two synthetic models (on the 40 subjects from HCP): **SynthFSL**, trained on labels obtained with FSL, and **SynthFree**, trained with FreeS labels. We also included **SynthSeg** (Billot et al., 2023) in our benchmark. This model was trained on labels obtained with FreeS from more than a thousand subjects (in 3 different dataset). These synthetic models were compared to classical models trained on real data: FastSurfer (**FastS**) which was trained on FreeS labels (Henschel et al., 2022) and Assembly Net (**AssN**) trained on manually defined labels from 53 subjects (including the MICCAI dataset) (Coup et al., 2020).

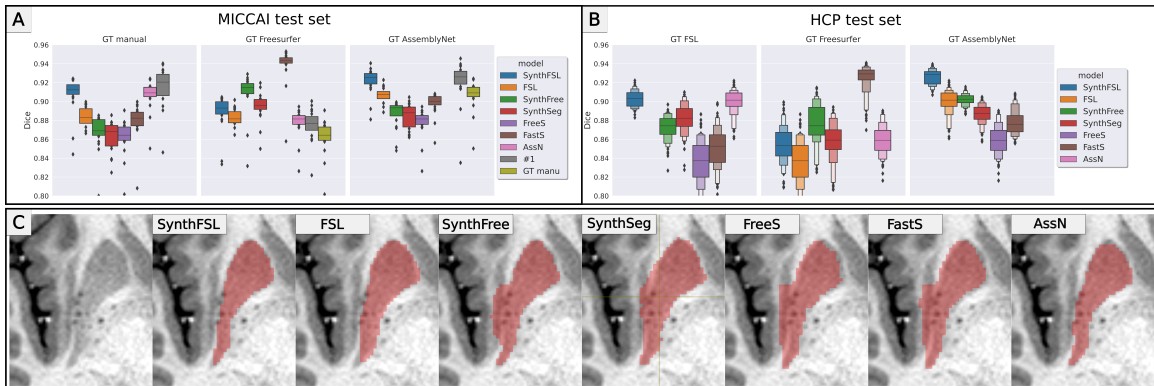

Figure 1: Dice score of Putamen for different models. A) 20 subjects from the MICCAI testset B) 80 subjects of the HCP testset. We change the segmentation chosen as Ground Truth respectively in column : manual segmentation or FSL, Freesurfer and AssemblyNet. C) Results for an axial slice of HCP Subject.

## 3. Results

The two widely used methods, FreeS and FSL, produce significantly divergent outcomes on both datasets. As illustrated in Fig.1, the SynthFSL model shows high performance on the MICCAI dataset when compared to manual segmentation. It has a similar DICE score as AssN despite the latter having been trained on real data with manual segmentation. On the other hand, using FreeS as ground truth will strongly degrade the performance across all models. The performance and ranking of the methods are highly consistent when evaluated on the HCP dataset. Despite a better image contrast in the Putamen, we observed large systematic errors for FreeS. As a consequence, the models trained with the labels from FreeS reproduce this bias. In particular, the predictions made by FastS closely align with those from FreeS, thereby replicating the same bias. Although SynthFSL was trained with synthetic data generated from FSL labels, its predictions are closer to AssN than to FSL, which supports the potential of synthetic models to mitigate inductive bias from the input labels. Nevertheless, synthetic models are clearly affected by variations in the definition of the label maps used for generating the training data: the predictions from SynthFree and SynthSeg are closer to FreeS than to SynthFSL. Note that this effect may be specific to putamen, given the large changes in its global shape induced by FreeSurfer systematic errors.

## 4. Discussion and Conclusion

The inductive bias in supervised learning is well documented, but is difficult to quantify and characterize. The empirical solution of designing unbiased manual annotation datasets is both challenging and resource demanding. Our results support the relevance of the synthetic learning approach for mitigating this problem. Previous publications (Billot et al., 2023; Valabregue et al., 2023) reported that the synthetic learning models do not perform as well as models trained on real data when evaluated using DICE scores. We argue that the difference is partly due to systematic bias present in the GT. Indeed when the manual GT is taken as reference, we observe very similar performance between the synthetic approach trained on the predictions from the best classical method on this dataset (SynthFSL) and a method trained on real data (AssN). On the other hand, the 5 dice point difference between SynthFree and FastFS when considering freesurfer as GT is an indirect measure of the systematic bias. Although we believe that the synthetic approach will help to reduce bias in the prediction, we show that the definition of the labels influences the results. It is then important to improve the anatomical accuracy of the labels used for generating synthetic images. The results from this study are specific to the Putamen structure, and further work is required to assess the generalization to other structures. Our observations should be also valid for any other kind of systematic bias. For instance, automated methods are known to be affected by variations in the image quality related to the acquisition settings (Hu et al., 2023). The current trend to improve robustness and generalization of segmentation models is to train on large multicentric datasets, using automated segmentation as GT (Svanera et al., 2024). Our study highlights the serious risk of obtaining models that reproduce the bias from the initial segmentation technique.

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
