# OpenReview forum: "Unraveling Systematic Biases in Brain Segmentation: Insights from Synthetic Training"
_MIDL.io/2024/Short_Papers — MIDL 2024 Short Papers_

### Official Review · Reviewer_sYfv · 2024-04-16

**Confidence:** 5
**Final Rating:** 4

**Review:**

Summary:
This paper evaluates the effect of using silver-standard ground truth for training and testing brain MRI segmentation models. This type of analysis is much needed in our field, since a lot of methods started to take liberties with respect to the quality of their GT. Here, the paper seems to be claiming that training models with synthetic images built from automated GT is fine, but that evaluating on automated GT is misleading.

Weaknesses:
- My main concern is that the paper lacks of a clear ditinction between the use of automated GT for either training or testing. At the moment, the result and discussion sections keep jumping from one to the other. I think these sections should have 1 separate paragraph for each use of automated GT.
- Section 2 is too compact and a bit confusing, especially regarding the separation between the two datasets. Also, baselines should be presented in a separate paragraph.
- "Billot et al., 2023 reported that the synthetic learning models do not perform as well as models trained on real data when evaluated using DICE scores." I don't think this is a claim of this paper. They show that models trained on real data perform better than SynthSeg when tested on the training domain, but that SynthSeg generalises much better to other domains. Please rephrase.
-There are a lot of typos (some of which are listed below). Please carefully proof-read the paper.

Minor:
- "These synthetic models", models are not synthetic, it's the data they were trained with. Please rephrase here and at other occurrences.
- Fig.1: I think the ordering of the method is confusing. It should follow the same logical order as in section 2. Also, it's strange not to use the same type of plots in 1.A and 1.B. Finally Fig.1 is not referenced in the text.
- "dice" has a capital D because it's named after a person
- please use consistent names for all methods throughout the paper. For example FreeSurfer is alternatively written as FreeS, freesurfer or FreeSurfer.
- Some letters (especially the "e" with accents) are missing in the author names in the references (for example Coupe et al.).

---

### Decision · Program_Chairs · 2024-04-26

Accept